# SARS-CoV-2 viral load as a predictor for disease severity in outpatients and hospitalised patients with COVID-19: A prospective cohort study

**Fredrikke Christie Knudtzen**[1,2,3,4]*, **Thøger Gorm Jensen**[3,5,6], **Susan Olaf Lindvig**[1], **Line Dahlerup Rasmussen**[1], **Lone Wulff Madsen**[1,2], **Silje Vermedal Hoegh**[5,6], **Malene Bek-Thomsen**[7], **Christian B. Laursen**[2,8], **Stig Lønberg Nielsen**[1©], **Isik Somuncu Johansen**[1,2©]

1 Department of Infectious Diseases, Odense University Hospital, Odense, Denmark, 2 Department of Clinical Research, University of Southern Denmark, Odense, Denmark, 3 Clinical Center of Emerging and Vector-borne Infections, Odense University Hospital, Odense, Denmark, 4 OPEN, Open Patient Data Explorative Network, Odense University Hospital, University of Southern Denmark, Odense, Denmark, 5 Department of Clinical Microbiology, Odense University Hospital, Odense, Denmark, 6 Research Unit for Clinical Microbiology, University of Southern Denmark, Odense, Denmark, 7 Department of Clinical Microbiology, Lillebælt Hospital, Vejle, Denmark, 8 Department of Respiratory Medicine, Odense University Hospital, Odense, Denmark

© These authors contributed equally to this work.
* fredrikke.christie.knudtzen@rsyd.dk

**Data Availability Statement:** Due to The Danish General Data Protection Regulation (GDPR), the data used in this article is not publicly available.

## Abstract

### Introduction

We aimed to examine if severe acute respiratory syndrome coronavirus 2 (SARS-CoV-2) polymerase chain reaction (PCR) cycle quantification ($C_q$) value, as a surrogate for SARS-CoV-2 viral load, could predict hospitalisation and disease severity in adult patients with coronavirus disease 2019 (COVID-19).

### Methods

We performed a prospective cohort study of adult patients with PCR positive SARS-CoV-2 airway samples including all out-patients registered at the Department of Infectious Diseases, Odense University Hospital (OUH) March 9-March 17 2020, and all hospitalised patients at OUH March 10-April 21 2020. To identify associations between $C_q$-values and a) hospital admission and b) a severe outcome, logistic regression analyses were used to compute odds ratios (OR) and 95% Confidence Intervals (CI), adjusting for confounding factors (aOR).

### Results

We included 87 non-hospitalised and 82 hospitalised patients. The median baseline $C_q$-value was 25.5 (interquartile range 22.3–29.0). We found a significant association between increasing $C_q$-value and hospital-admission in univariate analysis (OR 1.11, 95% CI 1.04–1.19). However, this was due to an association between time from symptom onset to testing and $C_q$-values, and no association was found in the adjusted analysis (aOR 1.08, 95% CI

Researchers can request access to the data from the Danish Patient Safety Authority (https:/stps.dk/en or by email stps@stps.dk) and the Danish Data Protection Agency (https://www.datatilsynet.dk/english/).

**Funding:** The authors received no specific funding for this work.

**Competing interests:** The authors have declared that no competing interests exist.

0.94–1.23). In hospitalised patients, a significant association between lower $C_q$-values and higher risk of severe disease was found (aOR 0.89, 95% CI 0.81–0.98), independent of timing of testing.

## Conclusions

SARS-CoV-2 PCR $C_q$-values in outpatients correlated with time after symptom onset, but was not a predictor of hospitalisation. However, in hospitalised patients lower $C_q$-values were associated with higher risk of severe disease.

## Introduction

As the novel Severe Acute Respiratory Syndrome Coronavirus-2 (SARS-CoV-2) sweeps through the world, detection of viral RNA by polymerase chain reaction (PCR) has become the gold standard for diagnosing coronavirus disease 2019 (COVID-19) [1, 2]. Nasopharyngeal or oropharyngeal swabs make up the majority of tests, since most patients are unable to produce sputum despite higher sensitivity of the latter [3].

In viral diseases, the PCR Quantification Cycle-value or Cycle threshold-value ($C_q$ or $C_t$-value) can be used as a surrogate for viral load, with inverse correlation between the $C_q$-value and viral load. The use of $C_q$-value as a prognostic marker for disease severity in viral respiratory infections has been tested with varying results [4–6]. For coronaviruses, there is evidence from cohort studies supporting a correlation between $C_q$-values in upper airway samples and disease severity for human coronavirus in children, and between upper airway sample viral loads and disease severity for SARS-CoV-1 and Middle East Respiratory Syndrome (MERS) coronavirus infections in adults [7–9].

The most common symptoms of COVID-19 are fever, cough and dyspnea, and the course of disease can be complicated with Acute Respiratory Distress Syndrome (ARDS), respiratory failure and death [10–12]. There is limited data on whether viral load of SARS-CoV-2 correlate with disease severity. Two Chinese studies found that both in- and outpatients with COVID-19 had lower $C_q$-values indicating higher viral loads early in their disease course [13, 14]. A German study including hospitalised patients diagnosed with COVID-19 found that viral loads were high in the initial oropharyngeal samples and declining in 1–2 weeks [15]. Hospitalised patients in China with severe disease were found to have higher initial viral loads and prolonged time to reach PCR-negativity compared with patients with mild disease [16, 17].

With this study, we aimed to examine if baseline PCR $C_q$-values can identify 1) SARS-CoV-2 positive patients at increased risk of hospitalisation, and 2) hospitalised COVID-19 patients at increased risk of severe disease.

We hypothesized that the initial PCR $C_q$-values were lower among hospitalised patients, as a surrogate for higher viral loads, compared with non-hospitalised patients. We also hypothesized that due to a failure to reduce viral burden after the initial infection phase, lower PCR $C_q$-values were related to severe disease in hospitalised patients.

## Materials and methods

### Study setting and population

Odense University Hospital (OUH) serves as a tertiary hospital for the Region of Southern Denmark (approximately 1.2 million inhabitants) as well as a secondary hospital for the island

of Funen (approximately 0.5 million inhabitants) [18]. The Danish public healthcare system supplies free, tax-funded healthcare for all residents.

Initially, the Danish national COVID-19 strategy was based on containment, where individuals who met the case definition were tested for SARS-CoV-2. The strategy later changed to mitigation, where only patients with symptoms of COVID-19 requiring hospital admission were tested for SARS-CoV-2.

## Data sources

We used the unique 10-digit personal identification number assigned to all individuals in Denmark at birth or upon immigration to link the following two registries electronically with laboratory data:

1. *The COVID-19 Hospital Cohort at OUH*; a prospective hospital-based cohort of all adult (≥18 years old) COVID-19 patients admitted or referred to OUH since March 10, 2020. The cohort is ongoing and consecutively includes patients diagnosed with COVID-19. All patients admitted until April 21, with an available PCR $C_q$-value were included in this study. More details about this cohort is published elsewhere [11].

2. *The COVID-19 Outpatient Cohort in the Region of Southern Denmark*; a database of all adult COVID-19 patients from the Region of Southern Denmark tested positive for SARS-CoV-2 between March 9, 2020 and March 17, 2020, who had an available PCR $C_q$-value and were not admitted to a COVID-19 unit during their course of disease.

## Data collection

For the hospital cohort, demography-, clinical-, laboratory-, management- and outcome data were gathered through review of medical records [11]. For the outpatient cohort, all eligible patients were invited to participate in an online survey two months after symptom onset. By signing an electronic consent form, a survey could be filled-out and electronically retrieved into a database. The data included information on demography, disease exposure, clinical symptoms of COVID-19, days until recovery and remaining symptoms (see **S1 Appendix**).

Data on PCR assays, type of airway samples (naso- and/or oropharyngeal swab, or sputum), PCR $C_q$-values and sample dates were collected from the Department of Clinical Microbiology, OUH and the Department of Clinical Microbiology, Lillebaelt Hospital.

## SARS-CoV2 PCR assays

SARS-CoV-2 detection was established on three different analysis platforms—the fully automated high throughput Cobas 6800 (Roche), the commercially available kit RealStar® SARS-CoV-2 RT-PCR kit 1.0 (Altona Diagnostics) and a laboratory developed real-time (RT)-PCR.

On Cobas 6800 a 650 μl respiratory sample (oropharyngeal swab sample or sputum) was applied onto the system and subsequently RNA extraction, reverse transcription, PCR analysis and detection were performed. SARS-CoV-2 detection on Cobas 6800 included an internal RNA control, primers and probes targeting the ORF1a/b non-structural region that is unique for SARS-CoV-2 (target 1) and a conserved region in the structural protein envelope E gene that is shared by the Sarbecovirus subgenus (target 2).

RNA used for the RealStar® SARS-CoV-2 RT-PCR test (Altona Diagnostics) and for the laboratory developed test was either extracted from: **1)** 500 μl respiratory sample material (oropharyngeal swab sample or sputum) using MagNA Pure 96 (Roche) with the extraction

kit DNA and viral NA large volume kit (Roche) using the protocol Pathogen Universal, or **2)** 300 μl respiratory sample material (naso- and oropharyngeal swab sample or sputum) using the Maxwell® 16 Viral Total Nucleic Acid Purification Kit (Promega) following the manufacturer's protocol. RealStar® SARS-CoV-2 RT-PCR kit 1.0 included three PCR analyses for the qualitative detection of and differentiation of Sarbecovirus subgenus (E gene) and SARS-CoV-2 specific RNA (S gene) in addition to an internal control. The kit was used according to the manufacturer's instructions with 30 μl reaction volume, and the 1-step RT-PCR was performed using Lightcycler 480 (Roche) or Stratagene Mx3005P (Agilent) in 96 well formats.

The laboratory developed real-time PCR E gene assay used for SARS-CoV-2 detection has been described previously [19]. This assay targeted a conserved sequence in the E gene region that is shared by the Sarbecovirus subgenus group (FP: `ACAGGTACGTTAATAGTTAATAG CGT`, RP: `ATATTGCAGCAGTACGCACACA`, Probe: `FAM-ACACTAGCCATCCTTACTGCGC TTCG-BHQ1`). Real-time PCR was performed in 15 μl reactions containing 3.75 μl TaqMan Fast Virus 1-Step master mix (ThermoFisher) with 1000 nM of each primer and 200 nM of the probe, and 5 μl RNA eluate. An internal RNA control (Newcastle disease virus vaccine strain; MSD) was added to the sample prior to RNA extraction (NDV-FWD-2: `5'-CACTGTCGG CATTATCGATGA-3'`, NDV-REV: `5'-GAGCATCGCAGCGGAAA-3'`, NDV-Probe: `5'-FA M-CCCAAGCGCGAGTTA-MGB-3'`). Reverse transcription and amplification was performed using Lightcycler 480 (Roche) in 384 well formats. The cycling conditions were as follows: Reverse transcription at 50°C for 5 min, inactivation of RT/initial denaturation at 95°C for 20 sec, followed by 45 cycles of 95°C for 15 sec, 60°C for 1 min for amplification.

For all assays, the PCR $C_q$-value cut-off for a negative test was set at 40 cycles.

As for choice of baseline test assay when more than one test was available, in assays with both a target for pan-Sarbecovirus and Coronavirus SARS-CoV-2 (Cobas 6800 and RealStar® SARS-CoV-2 RT-PCR test), the $C_q$-value for the pan-Sarbecovirus was chosen if available, if not the $C_q$-value for Coronavirus SARS-CoV-2 target was used. If one sample was tested with multiple assays, the assays were prioritized in the following order; 1. Cobas 6800 (Roche), 2. The laboratory developed real-time PCR, and 3. RealStar SARS-CoV-2 RT-PCR test (Altona Diagnostics). The order was chosen by an experienced molecular biologist and a senior clinical microbiologist. For both naso/oropharyngeal swabs and sputum samples, the baseline PCR-sample for each patient was set to the first registered test for that patient (= day 0). If there were multiple tests for one patient within the first 3 days (Day 0, 1 and 2), the sample with the lowest $C_q$-value within this period was chosen.

## Study design

We conducted a retrospective case-control study consisting of two different comparisons of sub-groups: **1)** a case-control study with the hospital cohort as cases and the outpatient cohort as controls, and **2)** a case-control study of our hospital cohort where the hospitalised patients with <u>severe disease</u> defined as ARDS, admittance to the Intensive Care Unit (ICU) and/or death during admission were included as cases, and the hospitalised patients with <u>moderate disease</u> (not fulfilling the definition of severe disease) were controls. The criteria for ARDS and grading of severity of ARDS were based on current international guidelines [20, 21].

## Exposures

PCR $C_q$-values were used to estimate predictors for **1)** hospital admission, and **2)** ARDS, ICU admission and/or death.

### Statistics

For baseline variables, descriptive statistics were reported as numbers and percentages for categorical variables and medians with interquartile ranges (IQR) for continuous variables. Chi-squared test or Fisher's exact test were used to compare categorical variables between groups, student's t-test and Wilcoxon Mann-Whitney test were used for parametric and non-parametric continuous variables, respectively.

We plotted the $C_q$-values according to days since symptom onset and examined a possible association using linear regression. To identify whether the $C_q$-value could predict **1)** hospital admission and **2)** ARDS, ICU admission and/or death, we used logistic regression to compute odds ratios (OR) and 95% confidence intervals (CI). Analyses were adjusted for potential confounding variables, which based on the current knowledge on COVID-19 was predetermined to be age, sex, comorbidities, Body Mass Index (BMI) and days from symptom debut to baseline PCR-sample (**1**). To reduce the risk of over-fitting, we only included confounders considered most important (sex, age) in the final multiple regression model **(2)**.

Data on all patients were registered in a REDCap database hosted by Open Patient data Explorative Network (OPEN) [22]. STATA version 15 (StataCorp LP, Texas) was used for data processing and analyses.

### Ethics approval

This study was registered as a quality development project at OUH, approved by the Danish Data Protection Agency (j.nr. 20/16169 and 20/20759) and the Danish Patient Safety Authority (Sagsnr. 31-1521-344). All data were handled in accordance with The General Data Protection Regulation (GDPR), the Danish Act on Data Protection, the Danish Act on Research Ethics Review of Health Research Projects and the Danish Health Act. The study adheres to the STROBE guidelines for observational studies. All patients gave informed consent for study participation prior to inclusion.

## Results

A total of 169 patients were included in the final cohort; 87 from the outpatient cohort (Fig 1).

### Patient characteristics

The baseline patient characteristics of the two cohorts are shown in Table 1.

The hospital cohort was significantly older (median age 63 years (IQR 55–74) vs 46 years (IQR 36–54), p<0.001), had higher weights (median BMI 26.5 (IQR 23.7–30.1) vs 24.6 (IQR 23.1–27.2), p = 0.003) and had a significantly higher proportion of all comorbidities except pulmonary diseases compared with the outpatient cohort.

### COVID-19 exposure and symptoms

Compared with the hospital cohort, the outpatients had a significantly higher degree of known exposure to COVID-19 (Table 1). In the outpatient group, 65 patients (77.4%) had travelled to a COVID-19 hot-spot in the 14 days prior to symptom onset. Of these, 58 patients (66.7%) had been on skiing holidays in the Tyrol region of Austria. COVID-19 symptoms in the two cohorts are illustrated in Fig 2.

Compared with the outpatient cohort, hospitalised patients more often had fever, cough, dyspnoea and gastrointestinal symptoms but less often rhinitis/throat pain and loss of smell/taste.

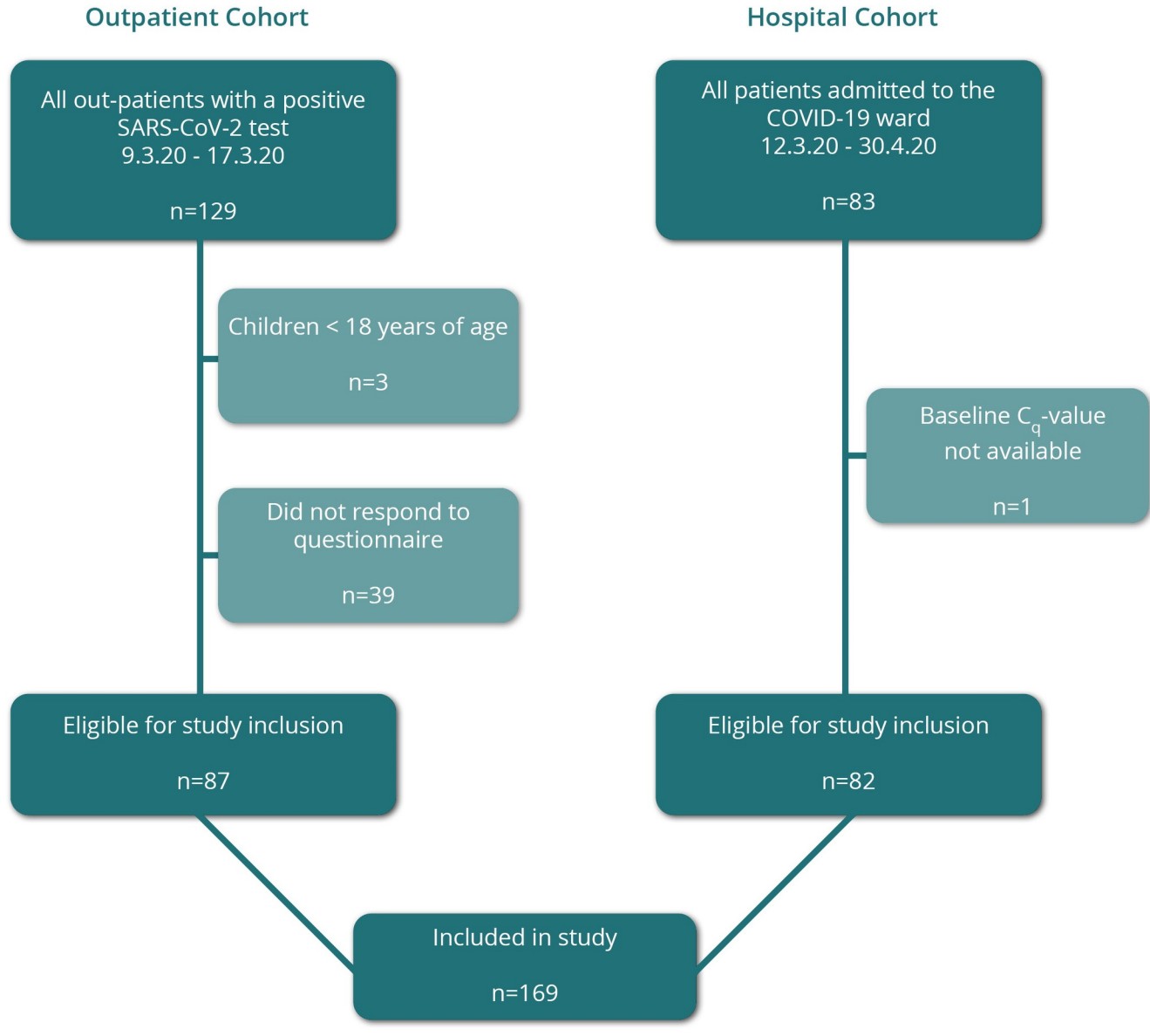

**Fig 1. Study inclusion of non-hospitalised and hospitalised patients into the Odense University Hospital COVID-19 cohort.**

### SARS-CoV-2 PCR $C_q$-value as a marker for hospital admission

The median baseline SARS-CoV-2 PCR $C_q$-value for the entire study population was 25.5 (IQR 22.3–29.0). The outpatients had a significantly lower median baseline SARS-CoV-2 PCR $C_q$-value (24.6, IQR 21.8–27.5) compared with the hospitalised patients (median $C_q$-value 26.9, IQR 23.6–31.3), p = 0.001 (Fig 3A).

We found a statistically significant association between an increasing baseline $C_q$-value and higher risk of admission to hospital (OR 1.11, 95%CI 1.04–1.19, p = 0.002) when using unadjusted logistic regression (see S1 Table). However, this was mainly due to a strong association between time from symptom onset and $C_q$-value (coefficient 0.26, 95%CI 0.15–0.38, p<0.001), as the patients in the outpatient cohort were tested significantly earlier in their course of disease compared with the hospital cohort (median 3 days (IQR 2–4) vs. 8 days (IQR 5–11),

**Table 1. Characteristics and exposures in a Danish outpatients and hospitalised patients with COVID-19.**

| Study population | | All patients n = 169 | Outpatient cohort n = 87 | Hospital cohort n = 82 | p-value |
|---|---|---|---|---|---|
| Age (years), median (IQR) N = 169 | | 54 (45–64) | 46 (36–54) | 63 (55–74) | <0.001 |
| Sex N = 169 | Male (%) | 110 (65.1) | 59 (67.8) | 51 (62.2) | 0.44 |
| BMI, median (IQR) N = 167 | | 25.6 (23.3–28.8) | 24.6 (23.1–27.2) | 26.5 (23.7–30.1) | 0.003 |
| Tobacco use (%) N = 167 | | | | | 0.28[a] |
| | Current smoker | 12 (7.2) | 7 (8.1) | 5 (6.2) | |
| | Former smoker | 54 (32.3) | 23 (26.7) | 31 (38.3) | |
| | Never smoker | 101 (60.5) | 56 (65.1) | 45 (55.6) | |
| Alcohol consumption (units/week) (%) N = 166 | | | | | 0.02[a] |
| | >7 for women / >14 for men | 18 (10.8) | 14 (16.5) | 4 (4.9) | |
| | ≤7 for women / ≤14 for men | 148 (89.2) | 71 (83.5) | 77 (95.1) | |
| Comorbidity (%) | | | | | |
| | Cardiovascular disease n = 169 | 25 (14.8) | 4 (4.6) | 21 (25.6) | <0.001 |
| | Hypertension n = 168 | 50 (29.8) | 15 (17.4) | 35 (42.7) | <0.001 |
| | Pulmonary disease n = 169 | 21 (12.4) | 8 (9.2) | 13 (15.9) | 0.19 |
| | Diabetes mellitus Type I+II n = 167 | 14 (8.4) | 1 (1.2) | 13 (15.9) | 0.001 |
| | Malignancy n = 167 | 19 (11.4) | 5 (5.9) | 14 (17.1) | 0.03 |
| Health care worker (%) N = 159 | | 16 (10.1) | 11 (12.9) | 5 (6.8) | 0.29 |
| **COVID-19 exposure** | | | | | |
| Travel to high risk area (%) N = 139 | | 78 (56.1) | 65 (77.4) | 13 (23.6) | <0.001 |
| | Austria (region of Tyrol) | 60 (76.9) | 58 (89.2) | 2 (15.4) | |
| | Italy | 5 (6.4) | 4 (6.2) | 1 (7.7) | |
| | Other | 13 (16.7) | 3 (4.6) | 10 (76.9) | |
| Contact with suspected/confirmed COVID-19 case (%) N = 169 | | 80 (47.3) | 50 (57.5) | 30 (36.6) | 0.007 |
| | Household | 26 (15.4) | 23 (26.4) | 3 (3.7) | <0.001 |
| | Colleague | 9 (5.3) | 7 (8.1) | 2 (2.4) | 0.17 |
| | Other | 37 (21.9) | 29 (33.3) | 8 (9.8) | <0.001 |

IQR = interquartile range; BMI = body mass index.

[a]Among all groups.

p<0.001). When adjusting for this difference in timing of testing, we no longer found a significant association between $C_q$-values and admission (OR 1.00, 95%CI 0.91–1.09, p = 0.97), irrespective of further adjustment for confounding factors (OR 1.08, 95%CI 0.94–1.24 p = 0.27).

## SARS-CoV-2 PCR $C_q$- values in different airway samples

We found no significant difference in median baseline $C_q$-values between naso-and/or-oropharyngeal swabs (143 patients; 13 naso-and-oropharyngeal and 130 oropharyngeal) and

**Fig 2. Number of patients displaying different symptoms among a non-hospitalised cohort (n = 87, displayed in dark green), and a hospitalised cohort (n = 82, displayed in light green) of adult COVID-19 patients.**

sputum samples (26 inpatients), with median $C_q$-values of 25.5 (IQR 22.3–28.8) and 24.4 (IQR 19.8–32.7), respectively (p = 0.61). Among the 165 patients with known symptom onset, we

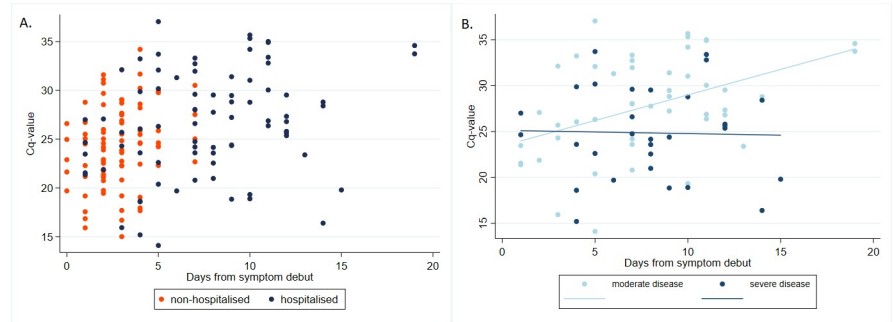

**Fig 3.** SARS-CoV-2 PCR baseline $C_q$-values and days from symptom debut to baseline sample in non-hospitalised (n = 87, in orange) and hospitalised (n = 82, in blue) patients with COVID-19 disease (**a**), and of the admitted patients with moderate (n = 51, light blue) and severe (n = 31, dark blue) disease (**b**).

observed a significantly shorter time from symptom onset to first PCR sample in patients tested with naso-and/or-oropharyngeal swabs compared with patients tested with sputum samples (median 3 days (IQR 2–7) vs 8 days (IQR 6–11), p<0.001).

## SARS-CoV-2 PCR $C_q$-value as a predictive marker for disease severity in hospitalised patients

A total of 31 of the 82 patients (38.0%) in the hospital cohort developed severe COVID-19 disease. Patients with moderate and severe disease did not differ with regards to sex, age, BMI, comorbidities, tobacco or alcohol consumption (see Table 2).

Patients with severe disease had significantly lower baseline $C_q$-values compared with patients with moderate disease (median 24.8 (IQR 21.0–28.8) vs 28.1 (IQR 24.3–33.2), p = 0.01). We found a statistically significant association between lower $C_q$-values and higher risk of severe disease (OR 0.89, 95%CI 0.81–0.98, p = 0.018). This association was independent of timing of the test in relation to symptom onset as well as presence of confounding factors including type of airway sample.

For patients with moderate disease, we found a direct linear association between the $C_q$-value and time of baseline test (Fig 3B). In contrast, we observed that patients with severe disease had a low baseline PCR $C_q$-value irrespective of time of testing. However, the regression coefficient between these two curves did not differ statistically (coef.-0.59 95%CI -1.20–0.02, p = 0.056) conferring to no significant interaction between $C_q$-value and time of test.

**Table 2. Characteristics of 82 patients admitted to Odense University Hospital with COVID-19, of which 31 patients had severe disease defined as either Acute Respiratory Distress Syndrome (ARDS), admittance to intensive care unit and/or death during admission, and 51 patients had moderate disease.**

| | | COVID-19 mild disease n = 51 | COVID-19 severe disease n = 31 | p-value |
|---|---|---|---|---|
| Age (years), median (IQR) N = 82 | | 61 (52–72) | 67 (58–78) | 0.09 |
| Sex N = 82 | Male (%) | 28 (54.9) | 23 (74.2) | 0.08 |
| BMI, median (IQR) N = 82 | | 26.1 (23.6–30.1) | 26.6 (23.7–32.2) | 0.73 |
| Tobacco use (%) N = 81 | | | | 0.24 |
| | Current smoker | 5 (9.8) | 0 (0.0) | |
| | Former smoker | 18 (35.3) | 13 (43.3) | |
| | Never smoker | 28 (54.9) | 17 (56.7) | |
| Alcohol consumption (units/week) (%) N = 81 | | | | 0.50 |
| | >7 for women / >14 for men | 3 (6.0) | 1 (3.2) | |
| | ≤7 for women / ≤14 for men | 47 (94.0) | 30 (96.8) | |
| Comorbidity (%) N = 82 | | | | |
| | Cardiovascular disease | 11 (21.6) | 10 (32.3) | 0.31 |
| | Hypertension | 18 (35.3) | 17 (54.8) | 0.08 |
| | Pulmonary disease | 8 (15.7) | 5 (16.1) | 0.96 |
| | Diabetes mellitus I+II | 7 (13.7) | 6 (19.4) | 0.54 |
| | Malignancy | 8 (15.7) | 6 (19.4) | 0.77 |

IQR = interquartile range; BMI = body mass index.

## Course of disease

Median symptom duration in the out-patient cohort was 11 days (IQR 5–16) when excluding fatigue and loss of taste/smell, which persisted two months after onset of COVID-19 disease in 15 (17.2%) and 27 patients (31.0%), respectively. A SARS-CoV-2 PCR test was repeated in 17 patients after a median of 8 days (IQR 6–9); all $C_q$-values increased, and 6 patients were PCR negative (Fig 4A).

In the hospital cohort, the median time from hospital admittance to either discharge (n = 78) or death (n = 4) was 7.5 days (IQR 3–11). Multiple PCR-samples were available in 33 patients with moderate disease and 18 patients with severe disease and showed a more complex pattern compared with the out-patient cohort. We observed a less linear increase in $C_q$-values, longer PCR positivity and several patients with subsequently decreasing $C_q$-values (Fig 4B+4C).

## Discussion

To our knowledge, this prospective study is the first to compare SARS-CoV-2 PCR $C_q$-values between non-hospitalised and hospitalised patients. Our most important findings were the strong linear association between $C_q$-values and time of testing after symptom onset, the correlation between lower $C_q$-values and increased disease severity in hospitalised patients and the lack of association between $C_q$-values and risk of hospitalisation.

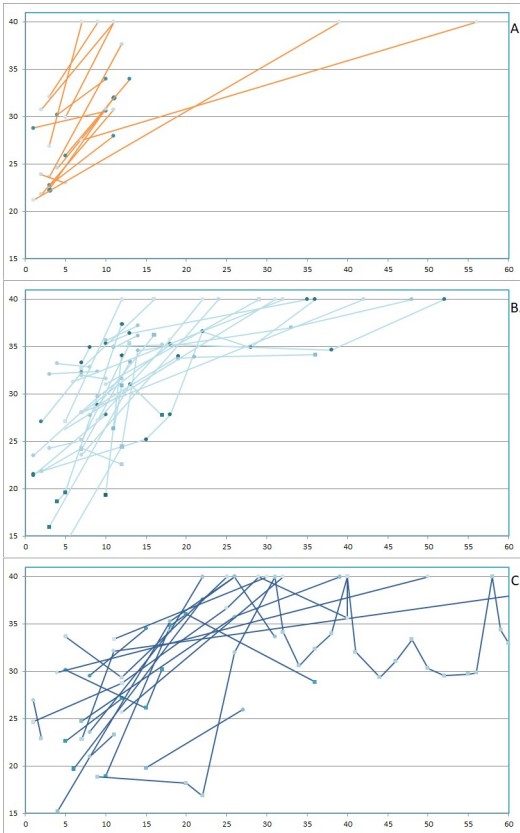

**Fig 4.** SARS-CoV-2 PCR $C_q$-values over time in 17 non-hospitalised patients (displayed in orange) (**a**), 33 hospitalised patients with moderate COVID-19 disease (displayed in light blue) (**b**) and 18 hospitalised patients with severe disease (displayed in dark blue) (**c**). The y-axis displays $C_q$-value and the x-axis displays days from symptom onset. Circle = naso-and/or-oropharyngeal swabs, squares = sputum samples.

Our results of a linear association between $C_q$-values and timing of the test after symptom onset are in line with available data that suggest higher SARS-CoV-2 viral loads in airway samples at symptom presentation followed by a gradual decrease [13–15, 17]. In this way, the novel Coronavirus differs from SARS-CoV-1, where viral loads were found to increase in airway samples until day 12–14 after symptom onset before decreasing [23, 24].

In hospitalised patients, we found that a lower $C_q$-value was associated with a significantly higher risk of severe disease irrespective of time of sampling and confounding factors. These findings are in line with the initial Chinese studies by Zheng and Liu, where patients with clinically severe disease had lower $C_q$-values and were PCR positive longer than patients with mild disease [16, 17]. Due to the limited size of our population, a specific PCR $C_q$-cutoff-value for patients in high risk of severe disease could not be estimated. Other studies are needed to explore this further in order to use it in a prediction model.

We could not confirm our hypothesis of an association between lower baseline $C_q$-values and higher risk of hospital admission when adjusting for timing of the test and confounding factors. To our knowledge, there are no available studies that have investigated this possible correlation.

Two systematic literature reviews regarding the use of PCR $C_q$-values in SARS-CoV-2 have been published since we undertook our study [25, 26]. In accordance with our findings, both studies report evidence of increasing $C_q$-values in respiratory samples over time, and an association between $C_q$-values and disease severity in hospitalized patients. However, the evidence was not conclusive and more data is needed in this area.

Of symptoms of COVID-19, we found significantly more patients in the hospitalized cohort with cough, dyspnea, fever and gastrointestinal symptoms. On the other hand, significantly more non-hospitalized patients suffered from rhinitis/throat pain and change in taste and/or smell. Other studies have shown conflicting results regarding change in taste/smell and severity of disease [27, 28].

The main strength of this study is the well-described cohort with near-complete data of high quality for all patients as well as electronically retrieved $C_q$-values. Furthermore, patients in both cohorts have been tested for SARS-CoV-2 based on the national standardized guidelines.

Our study has some limitations. Available data show varying inter-test agreement between different SARS-CoV-2 assays, especially in samples with high $C_q$-values [29–32]. In our study, three different PCR assays were used. This may have affected the reproducibility of the results. The different assays used reflect time and availability of assays during the pandemic; in the beginning most patients were tested using the in-house Flow, which was later replaced by the Cobas 6800. We also included results from both naso- and/or-oropharyngeal swabs and sputum samples, the latter only from hospitalised patients. Though we did not find any significant difference between sputum and oropharyngeal baseline $C_q$-values, this could be explained by the time of sampling, as the sputum samples were generally tested later in the patients' disease course. However, when including type of airway sample in our regression model for hospitalised patients, it did not alter the results. All airway-samples used in our study were sampled after clinical indication, and not as part of a research project. The airway swabs have therefore been sampled by different medical personnel. As this is an operator-dependent procedure, this lack of standardization may have affected the results. Whereas data on the hospital cohort was based on hospital files, data from the outpatient cohort was based on questionnaires filled out approximately two months after onset of disease. Therefore, recall bias cannot be excluded. Due to the small size of the two cohorts, we cannot exclude a risk of type 2 errors.

Finally, despite omission of confounding variables deemed not statistically significant, we cannot exclude some degree of over-fitting of the multivariate regression analyses. More

research in this area is needed, and larger cohorts would be able to confirm our findings with greater certainty.

There are still questions that need to be enlightened regarding why some patients get severe COVID-19 disease and others do not. Our findings suggest that clinicians cannot use the baseline $C_q$-value in outpatients to predict risk of hospitalisation later in their disease course. However, treating physicians should be vigilant of admitted patients with initial low $C_q$-values in their airway samples. When interpreting $C_q$-values, time of symptom onset should be considered, and patients with continuously low $C_q$-values should be closely monitored.

In conclusion, SARS-CoV-2 PCR $C_q$-values correlated with time after onset of symptoms. Early in the disease course $C_q$-values were low as a sign of high viral loads. We did not find $C_q$-values to be a predictor for hospitalisation. However, in hospitalised patients lower $C_q$-values were found to be predictive of more severe disease.

## Supporting information

**S1 Appendix. Questionnaire for the COVID-19 Outpatient Cohort–Region of Southern Denmark.**
(DOCX)

**S1 Table. Logistic regression model displaying univariate and multivariate estimates of risk factors associated with hospital admission in patients with SARS-Co-V-2.**
(DOCX)

## Acknowledgments

For the laboratory-developed real time PCR, the primer and probe sequences targeting the internal control virus were kindly provided by Dr. Kurt Handberg, Department of Clinical Microbiology, Aarhus University Hospital. We thank Benedicte Christie Knudtzen for help with graphic design of Figs 1 and 2.

## Author Contributions

**Conceptualization:** Fredrikke Christie Knudtzen, Thøger Gorm Jensen, Stig Lønberg Nielsen, Isik Somuncu Johansen.

**Data curation:** Fredrikke Christie Knudtzen, Thøger Gorm Jensen, Susan Olaf Lindvig, Line Dahlerup Rasmussen, Lone Wulff Madsen, Silje Vermedal Hoegh, Malene Bek-Thomsen, Christian B. Laursen, Stig Lønberg Nielsen, Isik Somuncu Johansen.

**Formal analysis:** Fredrikke Christie Knudtzen, Thøger Gorm Jensen, Line Dahlerup Rasmussen, Lone Wulff Madsen, Stig Lønberg Nielsen, Isik Somuncu Johansen.

**Investigation:** Fredrikke Christie Knudtzen.

**Methodology:** Fredrikke Christie Knudtzen, Line Dahlerup Rasmussen, Stig Lønberg Nielsen, Isik Somuncu Johansen.

**Project administration:** Fredrikke Christie Knudtzen.

**Supervision:** Isik Somuncu Johansen.

**Visualization:** Fredrikke Christie Knudtzen.

**Writing – original draft:** Fredrikke Christie Knudtzen, Thøger Gorm Jensen, Susan Olaf Lindvig, Line Dahlerup Rasmussen, Lone Wulff Madsen, Silje Vermedal Hoegh, Malene Bek-Thomsen, Christian B. Laursen, Stig Lønberg Nielsen, Isik Somuncu Johansen.

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
