## [Decision Letter · Decision Letter 0]

12 Jul 2021

PONE-D-21-10416

SARS-CoV-2 viral load as a predictor for disease severity in outpatients and hospitalised patients with COVID-19: a prospective cohort study

PLOS ONE

Dear Dr. Knudtzen,

Thank you for submitting your manuscript to PLOS ONE. After careful consideration, we feel that it has merit but does not fully meet PLOS ONE’s publication criteria as it currently stands. Therefore, we invite you to submit a revised version of the manuscript that addresses the points raised during the review process.

We look forward to receiving your revised manuscript.

Kind regards,

Giordano Madeddu

Academic Editor

PLOS ONE

2. Please ensure that you have provided the primer sequences for all assays performed in the study.

Additional Editor Comments (if provided):

Reviewers' comments:

Reviewer's Responses to Questions

**Comments to the Author**

1. Is the manuscript technically sound, and do the data support the conclusions?

Reviewer #1: Yes

Reviewer #2: Partly

2. Has the statistical analysis been performed appropriately and rigorously? 

Reviewer #1: Yes

Reviewer #2: Yes

3. Have the authors made all data underlying the findings in their manuscript fully available?

Reviewer #1: Yes

Reviewer #2: Yes

4. Is the manuscript presented in an intelligible fashion and written in standard English?

Reviewer #1: Yes

Reviewer #2: No

5. Review Comments to the Author

Reviewer #1: assessing the viral load in Sars Cov 2 infection is very important as a predictor in hospitalized patients to evaluate their clinical and therapeutic progress over time. this publication allows us to correctly evaluate the parameters taken into consideration

Reviewer #2: Introduction

The authors evaluated the severity of the diseases with the initial viral load. I suggest adding in the introduction the main symptoms of COVID-19 and the common complication. I suggest some paper that you could read and use for the introduction: https://doi.org/10.26355/eurrev_202007_22291;
https://doi.org/10.1371/journal.pone.0248009.

Methods

The study design is not clear. Therefore, I suggest re-writing it clearer (We conducted a retrospective case-control study […]).

The authors wrote that they used as cases hospitalized patients and outpatients as control, but in the next sentence, they wrote that severe hospitalized patients are the cases, and moderate hospitalized patients are the control.

Who performed the nasopharyngeal swabs? Was it always the same person? I am asking it because the swab is an operator-dependent procedure, and it could be an important bias.

Have you considered as severe cases all people who died? Or have you excluded non COVID-19 related deaths?

Results

I suggest adding a table with the logistic regression in order to make the result more clear. Furthermore, in my opinion, Table S2 should be added in the manuscript and not in the supplemental material.

It is not clear how they compare the different samples. In my opinion, the authors should compare only the samples coming from the same patients simultaneously; otherwise, too many biases are present. If it is not possible, I suggest removing this part.

Discussion

Some studies aimed that the presence of anosmia was correlated with a lower incidence of severe diseases. Other studies have not found this correlation. The authors could add a sentence about it considering the results. I suggest some paper that could be read and use: 10.1017/S0022215121001651; 10.1002/hed.26204

Figures

The quality of the figures is poor, and it is not easy to evaluate them. Please upload high-resolution figures, better in .pdf

Table 1.

Some percentages are missing.

The authors should add the captain explaining all abbreviations used in the tables.

I suggest removing from the Table’s title “Continuous variables are given as medians with interquartile ranges (IQR), categorical variables as percentages”.

6. PLOS authors have the option to publish the peer review history of their article (what does this mean?). If published, this will include your full peer review and any attached files.

Reviewer #1: No

Reviewer #2: No

---

## [Author Response · Author response to Decision Letter 0]

16 Aug 2021

Dear reviewers,

Thank you for taking the time to review our manuscript. We have altered the manuscript based on your constructive comments, and we believe the alterations have strengthened the manuscript. 

Here is a point-by-point response to the comments:

1. Is the manuscript technically sound, and do the data support the conclusions?

Reviewer #1: Yes

Reviewer #2: Partly

2. Has the statistical analysis been performed appropriately and rigorously?

Reviewer #1: Yes

Reviewer #2: Yes

3. Have the authors made all data underlying the findings in their manuscript fully available?

Reviewer #1: Yes

Reviewer #2: Yes

4. Is the manuscript presented in an intelligible fashion and written in standard English?

Reviewer #1: Yes

Reviewer #2: No

Reponse: the manuscript has been read and proofed by a native English speaker. 

5. Comments from Reviewers

Reviewer #1: assessing the viral load in Sars Cov 2 infection is very important as a predictor in hospitalized patients to evaluate their clinical and therapeutic progress over time. this publication allows us to correctly evaluate the parameters taken into consideration

Response: thank you for this feedback. We agree that we present data worth taken into consideration when interpreting viral loads in patients with SARS-CoV-2.

Reviewer #2: 

Introduction

The authors evaluated the severity of the diseases with the initial viral load. I suggest adding in the introduction the main symptoms of COVID-19 and the common complication. I suggest some paper that you could read and use for the introduction: https://doi.org/10.26355/eurrev_202007_22291;
https://doi.org/10.1371/journal.pone.0248009.

Response: we thank the reviewer for this suggestion, and we have added information about symptoms and complications of COVID-19 in the introduction. We have read the suggested articles with great interest, and we have added the references to the articles in the introduction. 

Methods

The study design is not clear. Therefore, I suggest re-writing it clearer (We conducted a retrospective case-control study […]). The authors wrote that they used as cases hospitalized patients and outpatients as control, but in the next sentence, they wrote that severe hospitalized patients are the cases, and moderate hospitalized patients are the control.

Response: we thank the reviewer for this relevant comment. Under the sub-headline “Study design” in the Methods section, we have altered the text accordingly, to make it clearer what the study design consists of, and that there are two different case-control sub-studies within this study, with different cases and control-groups. 

Who performed the nasopharyngeal swabs? Was it always the same person? I am asking it because the swab is an operator-dependent procedure, and it could be an important bias.

Response: we thank the reviewer for raising this relevant concern. As our study was retrospective, all swabs were performed on clinical indication by different medical personnel. We have added information about this to the Discussion section under the study limitations. 

Have you considered as severe cases all people who died? Or have you excluded non COVID-19 related deaths?

Response: We define “severe diseases” in the Methods section under the sub-headline “Study Design”. Severe cases are defined as either ARDS, admittance to intensive care unit and/or death during admission. The mortality rate in our hospitalized cohort was low (only four deaths, all due to COVID-19), and this low mortality is discussed in detail in another publication by our group (Madsen LW et al. Low mortality of hospitalized patients with COVID-19 in a tertiary Danish hospital setting. https://doi.org/10.1016/j.ijid.2020.10.018). No non-COVID-19 related deaths were found in our cohort, so no patients were excluded on this basis. 

Results

I suggest adding a table with the logistic regression in order to make the result more clear. 

Response: we thank you for this relevant comment. We have added a supplementary table displaying the univariate and multivariate logistic regression of risk factors associated with hospital admittance. 

Furthermore, in my opinion, Table S2 should be added in the manuscript and not in the supplemental material.

Response: we appreciate this suggestion, and the table has been added to the manuscript. 

It is not clear how they compare the different samples. In my opinion, the authors should compare only the samples coming from the same patients simultaneously; otherwise, too many biases are present. If it is not possible, I suggest removing this part

Response: in the Methods section, under the subline “Data collection” and “SARS-CoV-2 PCR assays” we have tried to explain both how the samples were taken, from whom, how they were compared, and how we included/excluded the different samples. As the main aim of this study was to compare all airway samples from different patients to see if they could determine hospital admission and severe disease, we chose to include all different airway samples available, to mimic the clinical reality we work in as medical doctors. We are aware that this complicates both the method and the results, but we believe this presents the most useful results. We have discussed this choice and the limitation it causes in the Discussion part of the paper. 

Discussion

Some studies aimed that the presence of anosmia was correlated with a lower incidence of severe diseases. Other studies have not found this correlation. The authors could add a sentence about it considering the results. I suggest some paper that could be read and use: 10.1017/S0022215121001651; 10.1002/hed.26204

Response: this is a very interesting issue regarding SARS-CoV-2. We have followed the reviewer’s suggestion and included a section in the Results about symptoms including change in taste/smell. We have included the references by Vaira et al here. 

Figures

The quality of the figures is poor, and it is not easy to evaluate them. Please upload high-resolution figures, better in .pdf

Response: We thank you for this valuable response. We have altered the figures according to the PLOS ONE instructions, where they have all been uploaded to the Preflight Analysis and Conversion Engine (PACE) digital diagnostic tool and altered to ensure they meet the PLOS quality requirements.

Table 1.

Some percentages are missing.

The authors should add the captain explaining all abbreviations used in the tables.

I suggest removing from the Table’s title “Continuous variables are given as medians with interquartile ranges (IQR), categorical variables as percentages”.

Response: thank you for these suggestions. Table 1 has been altered accordingly.

---

## [Decision Letter · Decision Letter 1]

28 Sep 2021

SARS-CoV-2 viral load as a predictor for disease severity in outpatients and hospitalised patients with COVID-19: a prospective cohort study

PONE-D-21-10416R1

Dear Dr. Knudtzen,

We’re pleased to inform you that your manuscript has been judged scientifically suitable for publication and will be formally accepted for publication once it meets all outstanding technical requirements.

Kind regards,

Giordano Madeddu

Academic Editor

PLOS ONE

Additional Editor Comments (optional):

Reviewers' comments:

Reviewer's Responses to Questions

**Comments to the Author**

1. If the authors have adequately addressed your comments raised in a previous round of review and you feel that this manuscript is now acceptable for publication, you may indicate that here to bypass the “Comments to the Author” section, enter your conflict of interest statement in the “Confidential to Editor” section, and submit your "Accept" recommendation.

Reviewer #2: All comments have been addressed

2. Is the manuscript technically sound, and do the data support the conclusions?

Reviewer #2: Yes

3. Has the statistical analysis been performed appropriately and rigorously? 

Reviewer #2: Yes

4. Have the authors made all data underlying the findings in their manuscript fully available?

Reviewer #2: Yes

5. Is the manuscript presented in an intelligible fashion and written in standard English?

Reviewer #2: Yes

6. Review Comments to the Author

Reviewer #2: (No Response)

7. PLOS authors have the option to publish the peer review history of their article (what does this mean?). If published, this will include your full peer review and any attached files.

Reviewer #2: No

---

## [Editor Report · Acceptance letter]

4 Oct 2021

PONE-D-21-10416R1 

SARS-CoV-2 viral load as a predictor for disease severity in outpatients and hospitalised patients with COVID-19: a prospective cohort study 

Dear Dr. Knudtzen:

I'm pleased to inform you that your manuscript has been deemed suitable for publication in PLOS ONE. Congratulations! Your manuscript is now with our production department. 

Kind regards, 

on behalf of

Dr. Giordano Madeddu 

Academic Editor

PLOS ONE